A possible brachiosaurid (Dinosauria, Sauropoda) from the mid-Cretaceous of northeastern China

Liao Chun-Chi 1 2 3
Moore Andrew 4 andrew.j.moore@stonybrook.edu
Jin Changzhu 1 2
http://orcid.org/0000-0001-8236-1210 Yang Tzu-Ruei 1 2
http://orcid.org/0000-0002-1878-6430 Shibata Masateru 5 6
Jin Feng 7 8 9
Wang Bing 7 8 9
Jin Dongchun 7 8 9
Guo Yu 10
Xu Xing 1 2 xu.xing@ivpp.ac.cn
1 Key Laboratory of Vertebrate Evolution and Human Origins of Chinese Academy of Sciences, Institute of Vertebrate Paleontology and Paleoanthropology , Beijing , China
2 CAS Center for Excellence in Life and Paleoenvironment , Beijing, Beijing , China
3 University of Chinese Academy of Sciences , Beijing, Beijing , China
4 Department of Anatomical Sciences, Renaissance School of Medicine at Stony Brook University , NY , USA
5 Fukui Prefectural Dinosaur Museum , Fukui , Japan
6 Institute of Dinosaur Research, Fukui Prefectural University , Fukui , Japan
7 Yanji Municipal Bureau of Land and Resources , Yanji , China
8 Yanji Paleontological Research Centre , Yanji , China
9 Yanji Dinosaur Museum , Yanji , China
10 The Geological Museum of China , Beijing, Beijing , China
Farke Andrew
Electronic publication date: 2021 Aug 20
Publication date: 2021
Volume: 9
Electronic Location ID: e11957
Received 2021 May 14; Accepted 2021 Jul 21
Copyright: © 2021 Liao et al.
Copyright year: 2021
Copyright holder: Liao et al.
License: This is an open access article distributed under the terms of the Creative Commons Attribution License, which permits unrestricted use, distribution, reproduction and adaptation in any medium and for any purpose provided that it is properly attributed. For attribution, the original author(s), title, publication source (PeerJ) and either DOI or URL of the article must be cited.
License URL: https://creativecommons.org/licenses/by/4.0/

Keywords: Albian-Cenomanian, Early cretaceous, Brachiosauridae, Paleobiogeography, Cranial evolution, Longshan fauna, Longjing formation

Funding: National Natural Science Foundation of China 41688103 Yanji Dinosaur Research Center University of Chinese Academy of Sciences (UCAS) This work was supported by the National Natural Science Foundation of China (41688103), the Yanji Dinosaur Research Center, and a University of Chinese Academy of Sciences (UCAS) scholarship to Chun-Chi Liao. The funders had no role in study design, data collection and analysis, decision to publish, or preparation of the manuscript.

==============================
Brachiosauridae is a lineage of titanosauriform sauropods that includes some of the most iconic non-avian dinosaurs. Undisputed brachiosaurid fossils are known from the Late Jurassic through the Early Cretaceous of North America, Africa, and Europe, but proposed occurrences outside this range have proven controversial. Despite occasional suggestions that brachiosaurids dispersed into Asia, to date no fossils have provided convincing evidence for a pan-Laurasian distribution for the clade, and the failure to discover brachiosaurid fossils in the well-sampled sauropod-bearing horizons of the Early Cretaceous of Asia has been taken to evidence their genuine absence from the continent. Here we report on an isolated sauropod maxilla from the middle Cretaceous (Albian–Cenomanian) Longjing Formation of the Yanji basin of northeast China. Although the specimen preserves limited morphological information, it exhibits axially twisted dentition, a shared derived trait otherwise known only in brachiosaurids. Referral of the specimen to the Brachiosauridae receives support from phylogenetic analysis under both equal and implied weights parsimony, providing the most convincing evidence to date that brachiosaurids dispersed into Asia at some point in their evolutionary history. Inclusion in our phylogenetic analyses of an isolated sauropod dentary from the same site, for which an association with the maxilla is possible but uncertain, does not substantively alter these results. We consider several paleobiogeographic scenarios that could account for the occurrence of a middle Cretaceous Asian brachiosaurid, including dispersal from either North America or Europe during the Early Cretaceous. The identification of a brachiosaurid in the Longshan fauna, and the paleobiogeographic histories that could account for its presence there, are hypotheses that can be tested with continued study and excavation of fossils from the Longjing Formation.

Introduction

Brachiosauridae is a clade of titanosauriform sauropods and one of the most iconic groups of non-avian dinosaurs, with well-known exemplars that include the Late Jurassic taxa Giraffatitan and Brachiosaurus. Although the in-group membership and inter-relationships of the clade remain a subject of continued debate (e.g., D’Emic, 2012; Mannion et al., 2013; Mannion, Allain & Moine, 2017; Carballido et al., 2015, 2020; D’Emic, Foreman & Jud, 2016; Royo-Torres et al., 2017), Brachiosauridae and slightly less inclusive subclades are readily diagnosed by a suite of characteristics from across the skeleton that are rare or absent among other lineages of sauropods (Wilson & Sereno, 1998; D’Emic, 2012; Mannion et al., 2013; Mannion, Allain & Moine, 2017). These include bauplan-defining traits, such as an elongate humerus that nearly equals or exceeds the length of the femur, as well as more subtle features of the skull and post-cranial skeleton, including axially twisted maxillary dentition, a small contribution of the ischium to the acetabulum, and a relatively broad proximal end of metacarpal III.

The oldest known brachiosaurid is from the Oxfordian of France (Lapparent, 1943; Mannion, Allain & Moine, 2017), and available body fossil evidence indicates that the lineage survived until the early Late Cretaceous, with the youngest brachiosaurids known from around the Albian/Cenomanian boundary (Chure et al., 2010; D’Emic, Foreman & Jud, 2016). During the Late Jurassic, brachiosaurids enjoyed a cosmopolitan distribution that included Africa (Janensch, 1914), North America (Riggs, 1903), Europe (Antunes & Mateus, 2003; Mannion et al., 2013; Mannion, Allain & Moine, 2017; Mocho, Royo-Torres & Ortega, 2017) and possibly South America (Rauhut, 2006; but see Mannion et al., 2013). Until recently, regional extinction of brachiosaurids across much of their Late Jurassic range was thought to have limited the group to North America during the Early Cretaceous (D’Emic, 2012; Mannion et al., 2013). However, the discovery of the brachiosaurid Soriatitan from the late Hauterivian–early Barremian of Spain (Royo-Torres et al., 2017), isolated vertebrae from the Berriasian–Hauterivian Kirkland Formation of South Africa referrable to Brachiosauridae (McPhee et al., 2016), isolated teeth showing axial twisting from the Early Cretaceous of Lebanon (Buffetaut et al., 2006; Mannion et al., 2013), and, potentially, the disputed brachiosaurid Padillasaurus from the Barremian of Colombia (Carballido et al., 2015, 2020; Mannion, Allain & Moine, 2017) reject the hypothesis that brachiosaurids suffered a major range contraction across the Jurassic/Cretaceous boundary. Although fossil evidence has occasionally been advanced to suggest that brachiosaurids dispersed into Asia (Lim, Martin & Baek, 2001; You & Li, 2009), the evidence underlying these claims has not held up to subsequent scrutiny (Barrett et al., 2002; Ksepka & Norell, 2010; Mannion, 2011; see “Discussion”). The failure to recover compelling evidence of brachiosaurids in the well-sampled sauropod-bearing horizons of the Early Cretaceous of Asia has been interpreted as a genuine indication that their Laurasian range was limited to Europe and North America (Ksepka & Norell, 2010; D’Emic, 2012; Mannion et al., 2013).

In our 2016 expeditions to the Longshan Beds of the Longjing Formation in Yanji City, Jilin Province, northeastern China, we discovered a mid-Cretaceous (Albian–Cenomanian) terrestrial fauna that has produced more than two hundred vertebrate fossils, including dinosaurians, crocodyliforms, and testudines. Sauropod dinosaurs represent the dominant group in the Longshan fauna (Jin et al., 2018) with more than 60 bones belonging to at least 14 individuals discovered so far. Most of these specimens were collected in the course of controlled excavations by our team, but some additional fossils were retrieved from heaps of excavated sediment from a nearby construction site. Among the latter are numerous dinosaurian and other unidentified teeth, a relatively complete crocodylian specimen, and an isolated sauropod maxilla and partial dentary. Because they were collected by our team after their exhumation, it is unknown whether the maxilla and dentary were preserved in association, and we conservatively consider them to belong to separate individuals. Whereas other sauropod fossils excavated from the Longshan beds either lack brachiosaurid synapomorphies (e.g., middle dorsal vertebrae with long, dorsoventrally short transverse processes; ratio of anteroposterior length of proximal plate of ischium to ischial proximodistal length <0.25) or bear features not known to occur in brachiosaurids (e.g. subcylindrical tooth crowns; bifurcated anterior dorsal neural spines), the isolated sauropod maxilla exhibits a mosaic combination of morphological features that suggest brachiosaurid affinities. Here we describe the morphology of the isolated maxilla, report on phylogenetic analyses that support its referral to Brachiosauridae, and discuss paleobiogeographic scenarios that could account for the occurrence of a middle Cretaceous Asian brachiosaurid. We also describe the isolated sauropod dentary from the same site, and discuss the effects that treating this specimen as belonging to the same individual as the isolated maxilla has on our results.

Materials & Methods

Systematic Paleontology

Dinosauria Owen, 1842

Saurischia Seeley, 1887

Sauropoda Marsh, 1878

Brachiosauridae Riggs, 1904

Brachiosauridae indet.

Material. YJDM 00008, a partial left maxilla with dentition in situ.

Locality and horizon. The Longshan locality (42°52′10.0″N, 129°29′28.1″E) is located south of Yanji City, Jilin Province (Fig. 1). The beds at the Longshan locality are part of the lower portion of the Longjing Formation, which conformably overlies the Dalazi Formation (Jin et al., 2018; Zhong et al., 2021). Paleontological and radiochronological data indicate an Albian to Cenomanian age for the Longjing Formation (Jin et al., 2018). The fossil-bearing site from which YJDM 00008 was recovered lies a short distance above a tuff layer from near base of the Longjing Formation that has recently been dated to 101.039 ± 0.061 Ma (Zhang et al., 2018; Zhong et al., 2021); this finding is consistent with other U-Pb radiochronological dating of the uppermost part of the Dalazi Formation to 105.14 ± 0.37 (Zhong et al., 2021). Thus, the Longshan section likely includes the Albian/Cenomanian boundary, and most of the Longjing Formation can be considered Cenomanian in age (Zhong et al., 2021).

Figure 1 Map showing the locality of YJDM 00008 & YJDM 00006 (red flag) in Jilin Province, China.

Description and comparisons

Description of YJDM 00008 was facilitated by X-ray computed tomography scanning of the specimen. The scan was performed using the 450 kV industrial X-ray computed tomography scanner (developed by the Institute of High Energy Physics, Chinese Academy of Sciences (CAS)) at the Key Laboratory of Vertebrate Evolution and Human Origins, CAS. The specimen was scanned with a beam energy of 430 kV and a flux of 1.5 mA at a resolution of 160 um per pixel using a 360° rotation with a step size of 0.25°. A total of 1,440 projections were reconstructed in a 2,048 × 2,048 matrix of 2,048 slices using a two-dimensional reconstruction software developed by the Institute of High Energy Physics, CAS (Wang et al., 2019). Data were output in the raw file format and imported into Mimics v.19.0 (Materialise, 2015, Leuven, Belgium) and Dragonfly v.2021.1.0.977 (Object Research Systems, Inc, 2021, Montreal, Canada) for analysis and visualization. Raw CT scan data for YJDM 00008 is available on MorphoSource (https://www.morphosource.org/concern/media/000361358?locale=en) to qualified researchers.

Axial twisting of the maxillary dentition in YJDM 00008 (see below) was visualized and measured digitally, as in D’Emic & Carrano (2020). To ensure accurate measurement of these angles for each tooth, the X, Y, and Z viewing planes were re-oriented in Dragonfly v.2021.1.0.977 so as to align with the mesiodistal, apicobasal, and labiolingual axes of the tooth, and the angle of twisting was measured across the entirety of the tooth crown. This approach was also used to confirm the absence of axially twisted dentition in several other eusauropod taxa for which CT scan data were available (Bellusaurus IVPP V17768.1; an indeterminate diplodocine USNM 2672; Camarasaurus CM 11338; Euhelopus PMU 24705/1a-b; an undescribed mamenchisaurid skull IVPP V27936; Sarmientosaurus MDT-PV 2).

Anatomical terminology for major components of the maxilla follows Wilson et al. (2016). We also coin a new term, internal antorbital fossa, for the fossa on the maxillary portion of the antorbital cavity that spans the medial surface of the narial process and the dorsal surface of the main body of the maxilla. Phylogenetic definitions used in this study are given in Table 1.

Table 1 Phylogenetic definitions.

Clade name/author	Definition	Reference	
Neosauropoda
Bonaparte (1986)	The least inclusive clade containing Saltasaurus loricatus and Diplodocus longus	Wilson & Sereno (1998)	
Diplodocoidea
Marsh (1884)	The most inclusive clade that includes Diplodocus longus but excludes Saltasaurus loricatus	Wilson & Sereno (1998)	
Diplodocimorpha
Calvo & Salgado (1995)	Diplodocus, Rebbachisaurus, their most recent common ancestor, and all of its descendents	Taylor & Naish (2005)	
Macronaria Wilson & Sereno (1998)	The most inclusive clade that includes Saltasaurus loricatus but excludes Diplodocus longus	Wilson & Sereno (1998)	
Titanosauriformes
Salgado, Coria & Calvo (1997)	The least inclusive clade including Brachiosaurus altithorax and Saltasaurus loricatus	Salgado, Coria & Calvo (1997)	
Brachiosauridae Riggs (1904)	The most inclusive clade that includes Brachiosaurus altithorax but excludes Saltasaurus loricatus	Wilson & Sereno (1998)	
Somphospondyli Wilson & Sereno (1998)	The most inclusive clade that includes Saltasaurus loricatus but excludes Brachiosaurus altithorax	Wilson & Sereno (1998); Upchurch, Barrett & Dodson (2004)	
Euhelopodidae
Romer (1956)	The most inclusive clade that includes Euhelopus zdanskyi but excludes Neuquensaurus australis	D’Emic (2012)	
Titanosauria
Bonaparte & Coria (1993)	The least inclusive clade that includes Andesaurus delgadoi and Saltasaurus loricatus	Wilson & Upchurch (2003)	
Lithostrotia Wilson & Upchurch (2003)	The least inclusive clade containing Malawisaurus dixeyi and Saltasaurus loricatus	Wilson & Upchurch (2003); Upchurch, Barrett & Dodson (2004)	

Maxilla. YJDM 00008 comprises a partial left maxilla with some replacement teeth in situ (Figs. 2–6). The preserved length of the maxilla is 28 cm, and we estimate the complete maxilla to have been about 30 cm long along its ventral margin. The maxilla is thus shorter than in Brachiosaurus and Giraffatitan (about 40 cm long; Janensch, 1935; Carpenter & Tidwell, 1998; D’Emic & Carrano, 2020) but longer than the maxilla of Euhelopus (approximately 18 cm long; Poropat & Kear, 2013), suggesting that the skull of YJDM 00008 is intermediate in size between Euhelopus and Giraffatitan.

Figure 2 (A) Photograph and (B) line drawing of YJDM 00008 in lateral view.

Abbreviation: amf, anterior maxillary foramen; amp, anteromedial process; nf, narial fossa; jp, jugal process; sf, subnarial foramen; nvg, neurovascular groove. Scale bar equals 5 cm.

Figure 3 (A) Photograph and (B) line drawing of YJDM 00008 in medial view.

Abbreviation: amp, anteromedial process; ?a.pal, ?articular surface for the palatine; g.pm, groove for articulation with the premaxilla ; inaof, internal antorbital fossa; jp, jugal process; rf, replacement foramen; rt, replacement teeth; sf, subnarial foramen. Scale bar equals 5 cm.

Figure 4 Photograph and line drawing of YJDM 00008 in ventral view (A), (C) and in anterolateral view (B), (D).

Abbreviation: amp, anteromedial process; g.pm, groove for articulation with the premaxilla; inaof, internal antorbital fossa; jp, jugal process; nf, narial fossa; nvg, neurovascular groove; rf, replacement foramen; rt, replacement teeth; sf, subnarial foramen. Scale bars equal 5 cm.

Figure 5 Photograph and CT slices of the dentition of YJDM 00008 in lingual view (A, B), posterior view (C), and ventral view (D).

The red arrows indicate the second generation of replacement teeth. Scale bar for (A) equals 3 cm.

Figure 6 Successive CT slices demonstrating that the mesiodistal axis of the maxillary teeth (yellow bars) of YJDM 00008 are twisted longitudinally.

The pictures from top to bottom are cross-sections from dorsal to ventral.

The partial left maxilla of YJDM 00008 can be broadly divided into two parts: a relatively thick, dentigerous ventral portion, and a more delicately constructed dorsal portion. The former part of the maxilla is largely intact, except for the missing posteriormost and ventral parts of the maxillary body, including that part that would have articulated with the jugal (Figs. 2–3). The dorsal part of the maxilla is more fragmentarily preserved, and is missing the narial (=ascending; posterodorsal) process and the lateral surface of the maxilla in the region of the antorbital fenestra, the margins of which are not intact. The dorsal part of the maxilla has also suffered some taphonomic distortion, such that this region bows outward and overhangs the lateral surface of the ventral, dentigerous portion of the maxilla.

Externally, the dorsal portion of the maxilla exhibits a slight concavity, bounded anteroventrally by a crescentic rim, that demarcates the anterior end of the narial fossa. At its anterior extreme, the narial fossa is pierced by a large foramen that we interpret to be the anterior maxillary foramen. An anterolaterally positioned narial fossa is also seen in Camarasaurus (Madsen, McIntosh & Berman, 1995), Euhelopus (Poropat & Kear, 2013) and Brachiosauridae (e.g. Brachiosaurus, Carpenter & Tidwell, 1998; Giraffatitan, Janensch, 1935; Europasaurus, Sander et al., 2006; Marpmann et al., 2015; Abydosaurus, Chure et al., 2010), unlike in late-branching titanosauriforms and diplodocoids, in which the naris and the narial fossa are more posterodorsally positioned on the maxilla and located on the top of the skull (Curry Rogers & Forster, 2004; Whitlock, 2011a; Zaher et al., 2011; Tschopp & Mateus, 2017). Anterodorsally, the maxilla bears an elongate sulcus that would have accommodated the narial (=ascending) process of the premaxilla. A stout anteromedial (=premaxillary; anterodorsal) process projects from the maxilla immediately ventromedial to this sulcus. The anteromedial process articulated with the premaxilla, and in life would have received a posteromedially-directed process from the latter bone, for which it bears a groove on its dorsal surface. At the base of the anteromedial process is a semi-circular notch that corresponds to the maxillary half of the subnarial foramen (Figs. 2–3), the other half of which would have been provided by a complementary notch in the premaxilla. The subnarial foramen appears to have been mediolaterally oriented and visible in lateral view, as in diplodocoids, late-branching titanosauriforms (Wilson et al., 2016), and Euhelopus (Poropat & Kear, 2013) but unlike the dorsal orientation of this foramen in neosauropods like Camarasaurus (CM 11338; Madsen, McIntosh & Berman, 1995) and Giraffatitan (Janensch, 1935; Madsen, McIntosh & Berman, 1995).

Although the posterior end of the maxillary main body is incomplete, it is clear that the specimen lacks the strongly tapering, posteriorly directed jugal process of some late-branching titanosauriforms (e.g., Rapetosaurus, Curry Rogers & Forster, 2004; Tapuiasaurus, Wilson et al., 2016). Instead, the specimen bears the plesiomorphically blocky posterior end of the maxilla that characterizes taxa such as Giraffatitan (MB.R.2180.2; Janensch, 1935), Euhelopus (Wilson & Upchurch, 2009; Poropat & Kear, 2013), and Sarmientosaurus (Martínez et al., 2016). Dorsally, the posterior end of the maxilla is marked by a trough, which provided entry into the dorsal alveolar canal for the maxillary vessels and dorsal alveolar nerve (White, 1958; Porter & Witmer, 2020). Posterior to the level of the last alveolus, the lateral wall of the dorsal alveolar canal has broken away, exposing the interior of the canal in lateral view. The presence of a preantorbital foramen/fenestra cannot be confirmed, as the relevant portion of the dorsal alveolar canal that would have given rise to it ventrolaterally is missing. It is noteworthy, however, that a broad, shallow fossa embays the lateral surface of the maxilla immediately ventral to the broken dorsal alveolar canal, as such a fossa is present in some taxa with well-developed preantorbital openings (e.g., Giraffatitan MB.R.2180.2; Tapuiasaurus Wilson et al., 2016). The presence of a continuous, plate-like wall of bone along the length of the palatal shelf (preserved intact or otherwise evidenced by a broken edge) suggests that the preantorbital opening, if present, was separated from the antorbital cavity, in contrast to the condition in various diplodocids (e.g., Galeamopus Tschopp & Mateus, 2017) and titanosaurians (e.g., Nemegtosaurus Wilson, 2005) in which the preantorbital opening is broadly continuous with the antorbital cavity.

On the medial surface of the maxilla, at the junction of its dorsal and ventral portions, is the internal antorbital fossa. This fossa is bounded ventrally by the palatal shelf and anteriorly and laterally by that portion of the maxilla that floors the narial fossa externally and gives rise to the narial process. The latter part of the maxilla is thin-walled and plate-like where it meets the palatal shelf. The lateral wall of the internal antorbital fossa meets the palatal shelf at an abrupt, approximately 90-degree angle. The sharp angulation of the ventrolateral boundary of the internal antorbital fossa contrasts with the relatively smooth transition between the ventral and lateral margins of the fossa in various other sauropods (e.g., Camarasaurus (Madsen, McIntosh & Berman, 1995, Figs. 9–11), Euhelopus (Poropat & Kear, 2013, Fig. 2), Rapetosaurus (Curry Rogers & Forster, 2004, Figs. 3–4)), though it is possible that the sharpness of this angle has been exaggerated by taphonomic distortion.

The anterior margin of the internal antorbital fossa extends close to the anterior one-third of the maxillary tooth row, which is also seen in Euhelopus (Wilson & Upchurch, 2009; Poropat & Kear, 2013). In Bellusaurus (Moore et al., 2018), Camarasaurus (Madsen, McIntosh & Berman, 1995), Brachiosauridae such as Brachiosaurus, Giraffatitan, and Abydosaurus (Janensch, 1935; Carpenter & Tidwell, 1998; Chure et al., 2010) and possibly Rapetosaurus (Curry Rogers & Forster, 2004), the anterior margin of the internal antorbital fossa only extends to roughly half the length of the tooth row. The medial view of the maxilla is poorly described or hard to observe in other taxa, and thus the relative anterior extent of the internal antorbital fossa is difficult to characterize more broadly. At the posteromedial end of the palatal shelf there is a rough area which might be the contact surface for the palatine.

In medial view, there are ten nearly complete alveoli with in situ dentition (Figs. 3, 5). The lateral and posterior walls of an additional alveolus are preserved at the anterior end of the maxilla, indicating a total of at least 11 maxillary alveoli. For comparison, Camarasaurus has 8–10 maxillary teeth (Madsen, McIntosh & Berman, 1995; Ikejiri, Tidwell & Trexler, 2005), the brachiosaurids Europasaurus (Marpmann et al., 2015), Abydosaurus (Chure et al., 2010), Giraffatitan (Janensch, 1935), and Brachiosaurus (Carpenter & Tidwell, 1998; D’Emic & Carrano, 2020) have 12–13, 10, 12, and 14 teeth, respectively, the early-branching titanosauriform Euhelopus has approximately 9 to 10 (Wilson & Upchurch, 2009; Poropat & Kear, 2013), the late-branching titanosauriform Tapuiasaurus (Zaher et al., 2011; Wilson et al., 2016) has 12, and the Asian late-branching titanosauriform taxa Nemegtosaurus and Quaesitosaurus have only 8 or 9 maxillary teeth (Kurzanov & Bannikov, 1983; Wilson, 2005). The maxillary teeth of the specimen are perpendicular to the ventral margin of the maxilla, as in Brachiosauridae and Camarasaurus (Janensch, 1935; Madsen, McIntosh & Berman, 1995; Carpenter & Tidwell, 1998; Sander et al., 2006; Marpmann et al., 2015) as well as late-branching titanosauriforms such as Tapuiasaurus, Nemegtosaurus, Quaesitosaurus and Rapetosaurus (Kurzanov & Bannikov, 1983; Curry Rogers & Forster, 2001; Curry Rogers & Forster, 2004; ; Wilson, 2005; Zaher et al., 2011; Wilson et al., 2016). By contrast, the maxillary teeth are anteroventrally oriented in Euhelopus (Wilson & Upchurch, 2009; Poropat & Kear, 2013), and a similar condition is also observed in Diplodocoidea, such as in Apatosaurus (Berman & McIntosh, 1978), Dicraeosaurus (Janensch, 1935), Kaatedocus (Tschopp & Mateus, 2013), Galeamopus (Tschopp & Mateus, 2017) and Diplodocus (Wilson & Sereno, 1998). Maxillary alveoli are approximately evenly spaced, with anterior alveoli slightly larger than posterior ones. The last 6 alveoli have an anteroposterior width of 10 to 15 mm, but the anterior 4 alveoli can reach 18 to 20 mm. Slightly above the alveoli (approximate 10 mm dorsal to them) on the medial side, there are a series of deep neurovascular foramina—the replacement foramina—through which the replacement teeth are visible.

Dentition. All functional teeth are missing from the specimen, and at least two generations of replacement teeth are preserved (Fig. 5). The teeth are positioned in the anterior half of the maxilla, though the anterior positioning of the teeth is not as extreme as in Diplodocoidea (Janensch, 1935; Wilson & Sereno, 1998; Berman & McIntosh, 1978; Sereno et al., 2007; Tschopp & Mateus, 2013, 2017). The anterior teeth are slightly larger than the posterior ones, and the teeth are curved lingually.

The maxillary teeth are parallel-sided in labial view, lacking the mesiodistal expansion of the crown that is plesiomorphic for sauropod dentition. The tooth crowns taper apically, and in cross-section are roughly D-shaped, with a strongly convex labial face, as in Brachiosauridae (Janensch, 1935; Carpenter & Tidwell, 1998; Sander et al., 2006; Chure et al., 2010; Marpmann et al., 2015), various early-branching somphospondylans (e.g., Bonaparte, González Riga & Apesteguía, 2006; Rose, 2007; Torcida Fernández-Baldor et al., 2017), and Euhelopodidae (Wilson & Upchurch, 2009; D’Emic, 2012; Poropat & Kear, 2013, and unlike the subcylindrical crowns found in later-branching somphospondylans and most Diplodocoidea (Berman & McIntosh, 1978; Kurzanov & Bannikov, 1983; Curry Rogers & Forster, 2001; Curry Rogers & Forster, 2004; Wilson, 2005; Whitlock, 2011a; Zaher et al., 2011). The lingual surface of the crown is subtly divided into two faces—a wider, slightly concave mesial face and a narrower, relatively flat distal face—that are gently offset from one another (Figs. 5–6). Where these two lingual faces meet there is a low, longitudinal bulge. We interpret this feature to be homologous to the apicobasal lingual ridge that is plesiomorphic for sauropods and is present in some brachiosaurids, Sauroposeidon, Astrophocaudia, Sibirotitan, Euhelopus, and most non-neosauropod sauropod teeth (Barrett et al., 2002; Rose, 2007; D’Emic, 2013; Mannion et al., 2013; Averianov et al., 2018), although its development in YJDM 00008 is markedly weaker than in most other taxa with a lingual ridge. The mesial and distal edges of the crown are relatively smooth compared to the wrinkled surface of the lingual and labial surfaces. The maxillary teeth are twisted axially (Fig. 6, Table 2), a feature that has been recovered as a unique synapomorphy of Brachiosauridae or a slightly less inclusive clade (e.g., D’Emic, 2012; Mannion et al., 2013; Mannion, Allain & Moine, 2017; Marpmann et al., 2015; D’Emic & Carrano, 2020). The slenderness index (SI; ratio of length of crown to mesiodistal width (Upchurch, 1998)) for YJDM 00008 ranges from approximately 2.30 to 3.56 (Table 2). This range is similar to that observed in various brachiosaurids (2.2–2.68 in Vouivria; 1.72–2.93 in Giraffatian; 2.4–3.25 in Abydosaurus; 2.3 in Soriatitan) (Chure et al., 2010; Mannion, Allain & Moine, 2017; Royo-Torres et al., 2017), Malawisaurus (3.3–3.5; Gomani, 2005), Sarmientosaurus (2.0–3.7; Martínez et al., 2016) and most non-lithostrotian somphospondylans, such as Europatitan (2.2; Torcida Fernández-Baldor et al., 2017), Euhelopus (2.2–3.3; Chure et al., 2010), Yongjinglong (1.65–3.93; Li et al., 2014), and Huabeisaurus (3.36–3.46; D’Emic et al., 2013). In Phuwiangosaurus and lithostrotians other than Malawisaurus, the teeth are markedly more slender than in YJDM 00008: for example, the SI exceeds 5 in Rapetosaurus (Curry Rogers & Forster, 2004) and Pitekunsaurus (Filippi & Garrido, 2008), and ranges from 3.3 to 6.3 in Phuwiangosaurus (Chure et al., 2010), from 4.1 to 5.9 in the upper teeth of Tapuiasaurus (Zaher et al., 2011; Wilson et al., 2016), and from 3.9–4.67 in Nemegtosaurus (Wilson, 2005).

Table 2 Replacement teeth measurements.

All measurements were taken digitally in Dragonfly v.2021.1.0.977 on the oldest generation of replacement tooth within a given alveolus. Because it was not possible to observe textural differences of the enamel that distinguish the root from the crown, measurements of apicobasal crown length are necessarily approximations that may slightly overestimate this length. Rt, replacement tooth of a given alveolus.

	Rt2	Rt3	Rt4	Rt5	Rt6	Rt7	Rt10	
Apicobasal crown length (mm)	28.83	23.55	28.30	24.63	30.44	24.98	20.10	
Crown width (mm)	10.81	10.23	10.74	9.31	8.55	9.51	8.43	
SI	2.67	2.30	2.64	2.65	3.56	2.63	2.38	
Twist angle (degrees)	56	58	55	45	40	29	55	

The CT scans show that two replacement teeth are present in each tooth socket (Figs. 5B–5D), as in Bellusaurus (Moore et al., 2018) and Brachiosaurus (D’Emic & Carrano, 2020). The younger generation of replacement teeth is distodorsal to and overlapped labially by the more mature generation. The crowns of the younger generation of replacement teeth are oriented mesioventrally (Figs. 5B–5D). Some other neosauropods exhibit greater numbers of replacement teeth. Among macronarians, Camarasaurus and the ‘Río Negro titanosaur’ possess three replacement teeth per alveolus (Coria & Chiappe, 2001; D’Emic et al., 2013). This condition differs from that of Diplodocoidea, which present a high tooth replacement rate and more generations of replacement teeth (e.g., five in Diplodocus; 10 in Nigersaurus) (Sereno & Wilson, 2005; D’Emic et al., 2013).

Systematic Paleontology

Dinosauria Owen, 1842

Saurischia Seeley, 1887

Sauropoda Marsh, 1878

Macronaria Wilson & Sereno, 1998

Macronaria indet.

Material. YJDM 00006, a fragmentary right dentary.

Locality and horizon. As for YJDM 00008 (see above).

Description and comparisons

YJDM 0006 comprises a fragmentary right dentary missing much of its anterior, dentigerous ramus (Fig. 7). The preserved portion bears four alveoli and corresponding replacement foramina. The dentary bifurcates posteriorly into posterodorsal and posteroventral processes. A roughened area for reception of the surangular marks the lateral surface of the posterodorsal process. A forked posteroventral process—a feature that characterizes Tapuiasaurus (Wilson et al., 2016) and brachiosaurids other than Europasaurus (Janensch, 1935; Chure et al., 2010; Marpmann et al., 2015; D’Emic & Carrano, 2020) and that results from the development of a small accessory process on the posteroventral process—appears to be absent in YJDM 0006. However, because the posterior portion of the posteroventral process and part of its dorsal margin are missing, it remains possible that the accessory process was present but relatively posteriorly positioned.

Figure 7 Photographs of YJDM 00006 in lateral view (A), medial view (B), and dorsomedial view (C).

Abbreviation: pdp, posterodorsal process; pvp, posteroventral process; rt, replacement teeth. Scale bar for (A, B) equals 5 cm, and for (C) equals 2 cm.

All functional teeth are missing, but three replacement teeth are preserved in situ and visible externally. One of the replacement teeth is in the last alveolus and the other two are in the penultimate alveolus, indicating at least two generations of replacement teeth. The dentary tooth crowns are parallel-sided, taper apically, and have a D-shaped cross-section, as in YJDM 00008 and in macronarians plesiomorphically. It is not possible to discern whether the dentary teeth bore denticles. The mesiodistal diameters of the dentary tooth crowns are notably smaller than those of roughly corresponding maxillary teeth in YJDM 00008: for posterior positions in each element, the dentary tooth is half as wide as the maxillary tooth (approximately 4 mm in the last two dentary replacement teeth vs. 8.43 mm in the tenth maxillary replacement tooth). An unequally-sized upper and lower dentition is a widely distributed feature among neosauropods (Chure et al., 2010; Mannion et al., 2013), including diplodocoids (e.g., Diplodocus Holland, 1924), the late-branching brachiosaurid Abydosaurus (Chure et al., 2010), and various somphospondylans (e.g. Euhelopus, Poropat & Kear, 2013; Sarmientosaurus, Martínez et al., 2016; Nemegtosaurus Wilson, 2005; Tapuiasaurus, Wilson et al., 2016). Thus, if the maxilla and dentary are hypothesized to belong to a single individual, then their disparate dentitions could be consistent with a wide array of phylogenetic positions within Neosauropoda. However, this possibility should be tempered by two cautionary points. First, although an unequally-sized upper and lower dentition occurs throughout Neosauropoda, this feature is nonetheless sparsely known, as it can only be confirmed in specimens with sufficient single-individual cranial material. Thus, its true distribution across Neosauropoda remains uncertain. Second, unerupted replacement teeth were still in the process of developing when the animal(s) died, and their measured mesiodistal diameters may not reflect the size ratios of the fully erupted, functional dentition, especially if the teeth being compared are measured at different stages of growth. This also bears consideration when noting that the size disparity between the maxillary and dentary replacement teeth of the Yanji cranial material is somewhat greater than that observed in various other taxa for which an association of upper and lower jaws is certain. Whereas the posterior replacement teeth in the Yanji maxilla are twice as wide as complementary teeth in the dentary, this ratio is lower for the functional teeth of other neosauropods with unequal upper and lower dentitions, including an indeterminate diplodocine (~1.4; USNM 2672), Abydosaurus (~1.3; DINO 17848), Euhelopus (~1.3; PMU 24705/1a-b), Sarmientosaurus (~1.7; Martínez et al., 2016), Tapuiasaurus (~1.5; Wilson et al., 2016), and Nemegtosaurus (~1.5; Wilson, 2005). Under the assumption that the maxilla and dentary belong to a single individual, the relatively large disparity in upper and lower tooth size in the Yanji cranial material could indicate that complementary replacement teeth are imperfect size proxies for mature teeth, or that this animal had a potentially autapomorphic degree of tooth size disparity. Alternatively, the dentary may belong to a smaller-bodied individual than the one represented by the maxilla, and may represent a different taxon. The available information does not allow us to distinguish between these possibilities.

Phylogenetic materials and methods

We tested the phylogenetic affinities of the Yanji cranial material using a morphological character matrix based on that of Poropat et al. (2021). Although we conservatively consider the maxilla (YJDM 00008) and dentary (YJDM 00006) to belong to separate individuals, we tested the effect that including these two specimens together as a single operational taxonomic unit (OTU) has on the results of our phylogenetic analyses. The maxilla (YJDM 00008) could be scored for 17 (3%) of the 552 characters in the Poropat et al. (2021) matrix. Inclusion of the dentary in the same OTU allowed two additional characters to be scored: character 103, concerning the forked posteroventral process of the dentary (scored as absent), and character 107, concerning unequally sized diameters of upper and lower dentition (scored as present). Although it is possible that a forked posteroventral process was present in the Yanji dentary (see above), we scored this feature as absent (as opposed to ‘?’) because such a score should provide a more stringent test of the possible brachiosaurid affinities of the Yanji cranial material, given that brachiosaurids other than Europasaurus possess this process.

In addition to the Yanji cranial material, we also added to the Poropat et al. (2021) matrix recently redescribed brachiosaurid cranial material from the Late Jurassic Morrison Formation of Garden Park, Colorado, USA (USNM 5730; Marsh, 1891; Carpenter & Tidwell, 1998; D’Emic & Carrano, 2020). In a series of preliminary analyses (not shown) in which USNM 5730 and the Brachiosaurus OTU of Poropat et al. (2021) were scored separately, USNM 5730 was consistently recovered as a brachiosaurid, under both equal and extended implied weights parsimony analysis and with and without the inclusion of the Yanji cranial material. In accord with the proposed existence of a single brachiosaurid species in the Morrison Formation (D’Emic & Carrano, 2020), we included USNM 5730 in the Brachiosaurus altithorax hypodigm for the phylogenetic analyses conducted here.

The final data matrix consisted of 552 characters scored for 126 OTUs (Supplemental I ) and was subjected to both equal weights (EW) and extended implied weights (EIW) parsimony analysis (Goloboff, 2014) in TNT v.1.5 (Goloboff & Catalano, 2016). We ran two separate versions of these analyses: one in which the maxilla was the sole representative of the Yanji cranial material, and another in which the dentary was included alongside the maxilla in a single OTU. Character ordering, taxon sampling, and down-weighting of homoplasy followed Poropat et al. (2021). Eighteen characters (11, 14, 15, 27, 40, 51, 104, 122, 147, 148, 195, 205, 259, 297, 426, 435, 472, 510) were treated as ordered. Ten unstable taxa (Astrophocaudia, Australodocus, Brontomerus, Fukuititan, Fusuisaurus, Liubangosaurus, Malarguesaurus, Mongolosaurus, Ruyangosaurus, and the ‘Cloverly titanosauriform’) were excluded a priori from the EW parsimony analysis; two of these (Ruyangosaurus and the ‘Cloverly titanosauriform’) were re-instated as active taxa for the EIW analysis. In the latter analysis, we applied a concavity constant (k) of nine.

For both EW and EIW analyses, we used ‘New Technology’ search algorithms to identify the set of most parsimonious trees (MPTs). Fifty search replications were used as a starting point for each hit, and were run until the best score was hit 10 times, using random and constraint sectorial searches under default settings, five ratchet iterations and five rounds of tree fusing per replicate (‘xmult = replications 50 hits 10 css rss ratchet 5 fuse 5’). The initial set of MPTs recovered by the analysis was subjected to an additional round of tree bisection and reconnection (TBR) branch swapping to exhaustively sample all equal-length trees. Alternative placements of the Yanji cranial material were identified using the resols command. Character support was assessed in TNT using the apo command and in Mesquite 3.61 (Maddison & Maddison, 2019).

Phylogenetic Results

Maxilla-only OTU. The EW parsimony analysis resulted in 318,737 trees of 2,665 steps (consistency index = 0.218; retention index = 0.398). The Yanji maxilla was found to be a neosauropod of unclear affinities: it is equally well-supported as 1) a non-diplodocimorph diplodocoid, 2) a well-nested brachiosaurid most closely related to a clade including Soriatitan, Venenosaurus, Cedarosaurus, and Abydosaurus, and 3) an early-branching euhelopodid (Fig. 8).

Figure 8 Strict consensus topology resulting from the maxilla-only phylogenetic analysis under equal weights parsimony.

Red dashed lines indicate equally parsimonious positions for YJDM 00008.

Parsimony analysis under EIW produced 2,520 trees of 115.80653 steps. The strict consensus of these trees resolves the Yanji maxilla as a brachiosaurid. Although the composition and early branching pattern of Brachiosauridae differ between the EW and EIW analyses, in both sets of MPTs the Yanji maxilla is part of a well-nested group comprising Soriatitan, Venenosaurus, Cedarosaurus, and Abydosaurus (Fig. 9).

Figure 9 Time-calibrated strict consensus tree resulting from the maxilla-only extended implied weights analysis demonstrating the relationships among neosauropods and the phylogenetic position of YJDM 00008.

Time-calibrated strict consensus tree resulting from the maxilla-only extended implied weights analysis demonstrating the relationships among neosauropods and the phylogenetic position of YJDM 00008.

Maxilla+dentary OTU. Inclusion of the dentary in the Yanji OTU did not affect tree length for the EW parsimony analysis, but produced many more MPTs (more than one million). Unlike the maxilla-only analysis, the maxilla+dentary OTU is not found to be a brachiosaurid (at least not in the one million MPTs that we collected), and is instead only recovered as either a non-diplodocimorph diplodocoid or an early-branching euhelopodid.

Parsimony analysis under EIW produced 1,890 trees of 115.84075 steps. The strict consensus of these trees is identical to that of the maxilla-only EIW analysis, except that the Yanji cranial material is recovered as an earlier-branching brachiosaurid, in one of three positions: as sister to Vouivria, one node stem-ward of Vouivria, or one node apical to Vouivria. This earlier-branching position results from scoring a forked posteroventral process of the dentary as absent, as this feature is a synapomorphy of the clade that includes Brachiosaurus, Giraffatitan, and Abydosaurus.

Discussion

Previous evidence for Asian brachiosaurids

Fossil evidence has occasionally been advanced to suggest the presence of brachiosaurids in the Late Jurassic or Early Cretaceous of Asia, but these hypothesized occurrences have either not held up to subsequent scrutiny, or at best provide only equivocal support for Asian brachiosaurids.

Based on pre-cladistic morphological comparisons emphasizing tooth crown shape, the Late Jurassic (Oxfordian) sauropod Bellusaurus, from the Shishugou Formation of northwest China, was initially assigned to its own subfamily (Bellusaurinae) within the Brachiosauridae, then considered part of the superfamily Bothrosauropodidea (Dong, 1990). Subsequent work has failed to support brachiosaurid kinship for Bellusaurus. Although the taxon may potentially represent a neosauropod (e.g., Upchurch, Barrett & Dodson, 2004; Carballido & Sander, 2014; Moore et al., 2018, 2020; but see, e.g., Wilson & Upchurch, 2009; Mo, 2013; Mannion et al., 2019b), no analysis has ever recovered Bellusaurus as a brachiosaurid, and Bellusaurus lacks many of the synapomorphies that unite Brachiosauridae and its subclades, including twisted maxillary dentition.

Similarities to Bellusaurus led Ye, Gao & Jiang (2005) to assign the Late Jurassic Daanosaurus, from the upper beds of the Shaximiao Formation, to the Brachiosauridae, within the subfamily Bellusaurinae. Daanosaurus has yet to be included in a phylogenetic analysis capable of testing its potential relationship to brachiosaurids; the only phylogenetic analysis to date to have included Daanosaurus exclusively sampled Middle–Late Jurassic Chinese sauropods, finding the taxon to be closely related to Mamenchisaurus (Li et al., 2011). The authors of this study did not report the matrix or the methods used in their analysis, and thus the character data in support of their phylogenetic conclusions are unclear. While the relationships of Daanosaurus remain obscure, none of the available evidence indicates a close relationship to brachiosaurids. Several characteristics (e.g., opisthocoelous posterior dorsal vertebrae; a tab-like interruption of the prezygodiapophyseal lamina in middle–posterior cervical vertebrae) suggest that Daanosaurus may be a mamenchisaurid (AJ Moore, 2015, personal observation; Mannion et al., 2013; Moore et al., 2020), although macronarian affinities have also been proposed (D’Emic, 2012).

An isolated tooth from the Early Cretaceous (Barremian–Aptian) Jinju Formation of South Korea was cited as the first evidence for Asian brachiosaurids on the basis of a chisel-like wear facet on its lingual surface (Lim, Martin & Baek, 2001). Subsequent consideration of the specimen by Barrett et al. (2002) disputed the presence of this form of wear facet and rejected its referral to Brachiosauridae, but concurred that the element likely belongs to an early-branching titanosauriform. Other isolated sauropod teeth from the Berriasian–Hauterivian (Barrett et al., 2002) and the Barremian (Saegusa & Tomida, 2011) of Japan exhibit a mosaic of features that has been considered potentially consistent with, but not diagnostic for, brachiosaurid affinities, although it should be noted that neither these teeth, nor the isolated tooth from the Jinju Formation, have been described as exhibiting axial twisting, the only unambiguous synapomorphy of brachiosaurid dentition.

The Chinese sauropod Qiaowanlong, comprising a partial postcranial skeleton from the late Early Cretaceous (Aptian) Xiagou Formation (You et al., 2018), was initially described as an Asian brachiosaurid based largely on comparisons to Sauroposeidon (then considered a brachiosaurid) (You & Li, 2009). However, the morphological basis for this referral evaporated when subjected to phylogenetic analysis by Ksepka & Norell (2010), who recovered Qiaowanlong as a somphospondylan, as have all subsequent authors (e.g., D’Emic, 2012; Mannion et al., 2013; Carballido et al., 2017; see also Mannion, 2011).

Thus, all previous fossil evidence has fallen shy of demonstrating the presence of brachiosaurids in Asia. As we elaborate in the following section, we consider YJDM 00008 to provide the most compelling evidence to date of an Asian brachiosaurid, while acknowledging that the fragmentary nature of the specimen requires that this hypothesis be treated cautiously, pending future discoveries in the Longjing Formation.

Phylogenetic affinities of the Yanji maxilla

In the discussion that follows, we focus on the results of the maxilla-only phylogenetic analyses. Although the results of the maxilla-only and maxilla+dentary analyses are not radically different, we nonetheless favor the former over the latter, for two reasons. First, because the maxilla and dentary were retrieved post-exhumation from heaps of excavated sediment that also included other vertebrate specimens (see above), any evidence for association between these two elements has been lost. The most conservative approach, therefore, is to treat them separately. Second, although inclusion of the dentary alongside the maxilla in a single OTU allows two additional characters to be scored, the scores for these characters are somewhat speculative, as discussed above. Given this uncertainty, scoring these characters is useful primarily to the extent that it incorporates the maximum possible character conflict that may exist for the OTU, and thus more stringently tests the hypothesis that the Yanji cranial material belongs to a brachiosaurid—a hypothesis that otherwise rests on a single character state (see below). Under EIW parsimony, our preferred mode of phylogenetic inference, the Yanji cranial material is found to be a brachiosaurid with or without inclusion of the dentary, indicating that the potential character conflict introduced by the dentary does not overwhelm the support for brachiosaurid affinities that is afforded by the maxilla. We thus focus on the results of our maxilla-only analyses, while acknowledging the inherent limitations of analyzing fragmentary fossils. Near the end of this section, we discuss additional caveats that attend interpretation of YJDM 00008 as a brachiosaurid.

Both the EW and EIW parsimony analyses agree that the Yanji maxilla belongs to a neosauropod. This identification is supported by the presence in YJDM 00008 of parallel-sided dentition (character 108), a feature that is resolved as a synapomorphy of Neosauropoda (EW) or Neosauropoda + (Camarasaurus + Lourinhasaurus) (EIW). The EW parsimony analysis provides equivocal support for the Yanji taxon as a brachiosaurid, a non-diplodocimorph diplodocoid, or a euhelopodid (Fig. 8). Character support for the latter two positions is limited to a single, homoplastically distributed feature: possession of a laterally-visible subnarial foramen (character 75). A laterally-visible subnarial foramen reflects the absence of a markedly depressed narial fossa and is plesiomorphic for Eusauropoda, present in Shunosaurus and secondarily reacquired in Euhelopus, lithostrotians other than Malawisaurus (= Nemegtosaurus, Rapetosaurus, and Tapuiasaurus), and either Diplodocimorpha or Diplodocoidea (depending on whether character optimization is assumed to occur under delayed or accelerated transformation, respectively).

While the lateral exposure of the subnarial foramen suggests possible diplodocoid affinities for YJDM 00008, numerous features, mostly of the dentition, exclude the specimen from Diplodocimorpha. These include a relatively smooth dentigerous portion of the lateral surface of the maxilla (character 288; this region is marked by deep, dorsoventrally elongate vascular grooves in diplodocimorphs and Nemegtosaurus); a Slenderness Index of <4.0 (character 11), D-shaped mid-crown cross-sections (character 109; these are cylindrical in diplodocimorphs and Titanosauria), tooth crowns with concave lingual surfaces (character 110; these are convex in diplodocimorphs, Titanosauria, Abydosaurus, and Phuwiangosaurus), an apicobasally-oriented lingual ridge (character 111; this is only very weakly developed in YJDM 00008 and is absent in Jobaria, diplodocimorphs, some brachiosaurids, and most somphospondylans), and fewer than three replacement teeth per alveolus (character 453). The absence of cranial material known for Amphicoelias or either species of Haplocanthosaurus allows the Yanji maxilla to be recovered in all possible positions available to a non-diplodocimorph diplodocoid (Fig. 8). Such a hypothesis for the Yanji maxilla would extend the temporal range of non-diplodocimorph diplodocoids by approximately 45 million years, and would indicate that a heretofore unsampled lineage of diplodocoids survived into the middle Cretaceous. Until recently, evidence for Asian diplodocoids was scant and controversial (Upchurch & Mannion, 2009; Whitlock, D’Emic & Wilson, 2011; Xu et al., 2018). The discovery of the early Middle Jurassic dicraeosaurid Lingwulong from China, the first definitive Asian diplodocoid and the oldest known neosauropod, indicates that diplodocoids dispersed into or originated from East Asia while Pangaea was a contiguous landmass (Xu et al., 2018), and may presage future discoveries of the group in Asia. Nevertheless, the lack of more compelling diplodocoid/diplodocimorph synapomorphies in the maxilla and dentition of YJDM 00008, the extreme temporal and phylogenetic remove between YJDM 00008 and Lingwulong, and the paucity of convincing evidence for diplodocoids in the Early Cretaceous of Asia make referral of YJDM 00008 to Diplodocoidea unlikely.

A hypothesis of euhelopodid affinities for the Yanji maxilla is more consistent with the known spatiotemporal ranges of neosauropod dinosaurs. Whereas no undisputed diplodocoids are presently known in the Early Cretaceous of Asia (Upchurch & Mannion, 2009; Whitlock, D’Emic & Wilson, 2011; Xu et al., 2018), numerous non-titanosaurian somphospondylan taxa have been recovered from this interval, with members of the Euhelopodidae—an East Asian radiation of somphospondylans—being particularly well-represented (D’Emic, 2012; Mannion et al., 2013, 2019a). Like the hypothesis of diplodocoid kinship, however, support for a position at the base of Euhelopodidae relies solely on the presence of a laterally-visible subnarial foramen, a homoplastically distributed feature that is thus far known only for the eponymous Euhelopus among euhelopodids. Recent comparative anatomical and phylogenetic work has called into question the macronarian affinities of Euhelopus (Moore et al., 2020), suggesting that phylogenetic results relying solely on features shared with that taxon should perhaps be treated cautiously.

A consideration of the evolutionary scenarios implied by competing topological positions of YJDM 00008 leads us to favor brachiosaurid affinities for the specimen. The EIW parsimony analysis and a subset of the MPTs from the EW analysis indicate that the Yanji taxon is a well-nested brachiosaurid. Support for brachiosaurid affinities for YJDM 00008 rests on a single feature—the presence of axially twisted maxillary teeth (character 114; Figs. 6–7)—which, under EW parsimony analysis, provides no more or less support for brachiosaurid affinities than a laterally visible subnarial foramen does for diplodocoid and euhelopodid kinship. Unlike a laterally visible subnarial foramen, however, twisted maxillary dentition is a characteristic that otherwise lacks homoplasy within Eusauropoda, and has been universally recovered as an unambiguous synapomorphy (sensu Tschopp, Mateus & Benson, 2015) of Brachiosauridae or a slightly less inclusive clade by previous authors (e.g., D’Emic, 2012; Mannion et al., 2013; Mannion, Allain & Moine, 2017; D’Emic, Foreman & Jud, 2016; Carballido et al., 2020). The high consistency of this character (CI = 1 in all previous analyses) accounts for why the EIW parsimony analysis favors only brachiosaurid affinities for YJDM 00008: parsimony under EIW weights characters in proportion to the homoplasy they incur on the trees being compared, and thus treats brachiosaurid kinship for YJDM 00008 as more parsimonious than either diplodocoid or euhelopodid affinities because such a relationship avoids homoplasy in a character that is otherwise perfectly hierarchical (i.e. twisted maxillary dentition), at the expense of adding a step to an unavoidably homoplasious character (i.e. laterally-visible subnarial foramen). We agree with the epistemological arguments in favor of such trade-offs (Goloboff, 1993), and in light of recent simulations showing that EIW outperforms EW parsimony (Goloboff, Torres & Arias, 2017), prefer the former over the latter as a mode of phylogenetic inference. In the absence of compelling character conflict with other brachiosaurids or evidence for a wider distribution of strongly (30–45°) twisted dentition outside of Brachiosauridae, we thus consider the available data to be most consistent with the hypothesis that YJDM 00008 is a brachiosaurid, diagnosed by a laterally visible subnarial foramen.

The nested position of YJDM 00008 among Cedarosaurus, Venenosaurus, Soriatitan, and Abydosaurus is supported by the absence of denticles in the dentition (character 113; observable only in the latter two taxa and YJDM 00008). Most eusauropods later-branching than Jobaria lack denticles. However, marginal enamel tuberosities were reacquired in brachiosaurids, where they are present in a grade that includes Europasaurus, Vouivria, Brachiosaurus, and Giraffatitan, and were secondarily lost in the subclade to which YJDM 00008 belongs. It should be noted, however, that at least some brachiosaurids, as well as some other sauropod taxa, appear to exhibit an uneven distribution of denticles between the upper and lower jaws. Replacement teeth preserved in the maxilla of Brachiosaurus lack denticles, whereas at least some of those in the dentary bear denticles on their mesial edge (D’Emic & Carrano, 2020), a pattern that also characterizes Bellusaurus (Moore et al., 2018) and Abrosaurus (Ouyang, 1989). Preservation of the visible replacement teeth in the Yanji dentary (YJDM 00006) is insufficient to determine whether denticles are present. Thus, it remains possible that the Yanji sauropod(s) bore denticles on the dentary teeth, though such a finding would not perturb support for brachiosaurid affinities.

The close relationship between YJDM 00008 and several late-branching brachiosaurids may also find support from the very weak development of an apicobasally oriented lingual ridge (character 111) in the teeth of YJDM 00008. This ridge is plesiomorphic for eusauropods (Barrett et al., 2002; Mannion et al., 2013) and is present in brachiosaurids such as Vouivria (Mannion, Allain & Moine, 2017) and Giraffatitan (Janensch, 1935–36), but is absent in Jobaria, Diplodocoidea/Diplodocimorpha, most somphospondylans, and the brachiosaurid subclade that includes Abydosaurus and Soriatitan. While the presence of a lingual ridge in YJDM 00008 excludes it in all MPTs from the Abydosaurus + Soriatitan clade, its subtle development in the specimen is potentially consistent with the progressive evolutionary loss of the lingual ridge in a subset of brachiosaurids.

Our interpretation of YJDM 00008 as a brachiosaurid is tempered by two important caveats. First, while current evidence indicates that axially twisted maxillary dentition is an unambiguous synapomorphy of a subclade of Brachiosauridae, very little is known about maxillary evolution in non-titanosaurian somphospondylans (and next to nothing, if Euhelopus lies outside of Macronaria; Moore et al., 2020). This knowledge gap leaves open the possibility that strongly twisted maxillary teeth in fact characterize a more inclusive grade of titanosauriforms or macronarians than the presently available fossil evidence would suggest. Slight axial twisting has been noted for the maxillary teeth of Europasaurus (Marpmann et al., 2015)—a taxon whose brachiosaurid and titanosauriform kinship remains a topic of controversy (Mannion, Allain & Moine, 2017; Carballido et al., 2020)—as well as for a handful of non-brachiosaurid titanosauriforms, including isolated teeth of Astrophocaudia (D’Emic, 2013) and distal maxillary teeth of Tapuiasaurus (Wilson et al., 2016). Considered together, these observations suggest that twisted dentition may be more broadly distributed within Macronaria than is presently appreciated, and underscore that additional materials from early-branching somphospondylans are needed in order to robustly test whether marked axial twisting (~30–45°) of the maxillary dentition indeed constitutes an unambiguous brachiosaurid synapomorphy.

Second, we have yet to identify any other clear evidence for brachiosaurids in the Longshan fauna, although it should be noted that our initial observations on the other sauropod fossils from the Longshan beds of the Longjing Formation are still very preliminary (being based on only a subset of the total material that has been excavated) and have not been incorporated into a phylogenetic analysis. Morphological details of these sauropod fossils are instead more consistent with a euhelopodid or early-branching titanosaurian identity, as indicated by such features as subcylindrical tooth crowns (character 109; a titanosaurian synapomorphy, also present in diplodocimorphs), bifurcated postaxial cervical and anterior dorsal neural spines (character 132; widely distributed in non-titanosauriform eusauropods, and present in most euhelopodids, early-branching titanosaurians, and Opisthocoelicaudia), and a scapula lacking a subtriangular process at both the posteroventral corner of the acromion and the anteroventral edge of the scapular blade (characters 215 and 216; both processes are absent in Jiangshanosaurus and Huabeisaurus, among other eusauropods). Although the phylogenetic affinities of other sauropod material from the Longshan beds do not bear directly on the identity of YJDM 00008, the possibility that the latter belongs to a brachiosaurid may become unlikely if all other material from the Longshan beds is eventually shown to belong to a single somphospondylan taxon. Ultimately, additional evidence for or against the presence of a brachiosaurid in the Longshan fauna, and other details on the taxonomic diversity of this assemblage, await further study of excavated specimens and future excavation in the Longjing Formation.

Paleobiogeographic implications of Asian brachiosaurids

Assuming brachiosaurid affinities for YJDM 00008, at least two scenarios can be posited to explain the occurrence of a middle Cretaceous Asian brachiosaurid. The first proposal interprets the presence of a brachiosaurid in the Longjing Formation as resulting from dispersal of a lineage of brachiosaurids into East Asia at some point in the Early Cretaceous (or possibly the Late Jurassic). The results of our maxilla-only phylogenetic analysis are most consistent with a close relationship between YJDM 00008 and North American brachiosaurids, and hence a North American origin for the lineage that gave rise to YJDM 00008. As discussed above, however, the character data supporting this inference are very limited, and the relationships of YJDM 00008 among brachiosaurids (or perhaps neosauropods more broadly) are likely to change with future discoveries. Here, we briefly consider alternative dispersal routes available to either North American or European ancestors of YJDM 00008; consideration of the latter possibility is warranted based on the presence of the Spanish brachiosaurid Soriatitan in the polytomy to which YJDM 00008 belongs, as well as other evidence for apparent interchange between the sauropod faunas of Europe and Asia in the Early Cretaceous (see below).

Current information is consistent with either North America or Europe as a potential source of Asian emigrants in the Early Cretaceous (Poropat et al., 2016; Xu et al., 2018; and references therein). Considerable biogeographic and phylogenetic evidence indicates a close relationship between Asian and North American faunas in the middle Cretaceous (e.g., Russell, 1993; Cifelli et al., 1997; Chinnery-Allgeier & Kirkland, 2010; D’Emic, Wilson & Thompson, 2010; Zanno & Makovicky, 2011; Farke et al., 2014; Brikiatis, 2016; Dunhill et al., 2016; Poropat et al., 2016; Ding et al., 2020). Trans-European dispersal cannot be ruled out as an explanation for faunal similarities between Asia and North America (e.g., Chinnery-Allgeier & Kirkland, 2010; Brikiatis, 2016; Ding et al., 2020); indeed, recent quantitative analyses of dinosaurian biogeography have emphasized Europe as a likely gateway between Asia, North America, and other landmasses in the Early Cretaceous (Dunhill et al., 2016; Ding et al., 2020), although Zanno & Makovicky (2011) argued that trans-European dispersal between Asia and North America at this time would have been complicated by the periodic development of various geographic barriers. An alternative hypothesis entails emplacement of a Bering land bridge between Asia and North America for at least part of the Albian (Russell, 1993; Cifelli et al., 1997; Zanno & Makovicky, 2011; Poropat et al., 2016). A direct Beringian connection has been invoked to explain apparent late Early Cretaceous dispersal events for tyrannosauroids (e.g., Zanno & Makovicky, 2011), therizinosaurians (e.g., Zanno, 2010), and neoceratopsians (e.g., Farke et al., 2014), among other vertebrate groups (but see Brikiatis, 2016 for an alternative view). Uncertainty about the timing and duration of a late Early Cretaceous Bering land bridge and the importance of Europe as an intermediate between North America and Asia notwithstanding (Brikiatis, 2016), the balance of evidence suggests that a Beringian connection existed within a timeframe that could explain the arrival of brachiosaurids in East Asia from North America by the Albian/Cenomanian boundary.

A European origin for Asian brachiosaurids is also possible, and receives support from biogeographic and paleogeographic studies. Taxonomic surveys and empirical paleobiogeographic analyses indicate substantial faunal exchange between Europe and Asia in the Early Cretaceous (e.g., Russell, 1993; Upchurch, Hunn & Norman, 2002; Chinnery-Allgeier & Kirkland, 2010; Dunhill et al., 2016; Ding et al., 2020). Periodic establishment of a Russian Basin/Turgai marine barrier would have impeded terrestrial dispersal between Europe and Central Asia in the late Berriasian–early Hauterivian and early Albian, but otherwise connections between these landmasses are thought to have existed for much of the Early Cretaceous (Poropat et al., 2016 and references therein), providing potential routes for an ancestral population of European brachiosaurids to disperse into East Asia. This scenario is consistent with other fossil evidence that indicates commingling of Asian and European sauropod faunas in the Early Cretaceous. Isolated teeth from the Barremian of Spain bearing a distolingual boss—a feature that is otherwise known only in some East Asian sauropods, including the Berriasian–Hauterivian Euhelopus (Wiman, 1929; Wilson, 2002; Barrett & Wang, 2007; Suteethorn et al., 2013; Moore et al., 2020)—would seem to suggest that a subclade of euhelopodids spread across both Asia and Europe in the Early Cretaceous (Canudo et al., 2002). Recently, the discovery of an isolated anterior caudal vertebra of a rebbachisaurid in the Turonian Bissekty Formation of Uzbekistan, as well as possible rebbachisaurid teeth from the same formation, have been interpreted as evidence for dispersal of European rebbachisaurids into Central Asia sometime between the Barremian and Turonian (Averianov & Sues, 2021). It should be noted, however, that the morphological basis for identifying the Bissekty Formation anterior caudal vertebra as a rebbachisaurid has been critically challenged by a reappraisal of the specimen by Lerzo, Carballido & Gallina (2021), who rejected rebbachisaurid affinities and provided evidence in support of a titanosaurian identity, a hypothesis also previously favored by Sues et al. (2015) and Averianov & Sues (2017). Regardless of the affinities of the Bissekty Formation specimen, the presence of a brachiosaurid in the Longjing Formation can be explained by the existence of plausible dispersal routes connecting East Asia to both Europe and North America during much of the Early Cretaceous.

The second biogeographic scenario suggests that brachiosaurids and other major neosauropod lineages were widely distributed across Pangaea, including East Asia, before the separation of Laurasia from Gondwana in the latter half of the Middle Jurassic and the isolation of East Asia from the rest of Laurasia from the Callovian–Tithonian (Poropat et al., 2016, Xu et al., 2018, and references therein). In this scenario, the occurrence of YJDM 00008 in the middle Cretaceous of northeast China reflects the persistence of brachiosaurids in Asia from the Middle Jurassic through the Early Cretaceous. The heretofore unrecognized presence of brachiosaurids in the region during this time would thus reflect biased sampling of the fossil record. Such a scenario seems unlikely, given that substantial prospecting in Middle–Late Jurassic and Early Cretaceous (particularly Barremian–Albian) strata of China has yielded a rich sauropod record (118 collections containing sauropod specimens, according to the Fossilworks Database, April 15, 2021) that, to date, appears to be wholly devoid of brachiosaurids. Nevertheless, the possibility that sampling biases have obscured the presence of an early-arriving lineage of Asian brachiosaurids should not be dismissed out of hand. Indeed, pervasive sampling artifacts may be necessary to explain the apparent absence of undisputed neosauropods from the well-sampled, sauropod-rich Middle–Late Jurassic horizons of the Junggar and Sichuan basins, given the recent discovery of the dicraeosaurid Lingwulong in older strata of north central China (Xu et al., 2018). Possible explanations for the scarcity of neosauropods (including brachiosaurids) in the Middle–Late Jurassic and of brachiosaurids in the Early Cretaceous of Asia include low abundance or diversity of these groups in their ecosystems, and failure to sample the preferred habitats in which these groups were more abundant (Whitlock, 2011b; Xu et al., 2018). These explanations have been proposed to account for the relatively low occurrence of brachiosaurids in dinosaur-bearing localities of the Morrison Formation (D’Emic & Carrano, 2020). Thus, irrespective of the series of events that might have brought a lineage of brachiosaurids to Asia, their extreme rarity in currently sampled Early Cretaceous dinosaur-bearing horizons may reflect the concerted effects of an overall low abundance and poor sampling of preferred habitats.

Conclusions

The recent discovery of a fossil-rich horizon near the base of the Albian–Cenomanian Longjing Formation has yielded numerous dinosaurian and other terrestrial vertebrate specimens, including an isolated maxilla of a neosauropod. Although fragmentary, this specimen preserves a striking morphology—axially twisted dentition—that is otherwise present only in brachiosaurids. Referral of YJDM 00008 to Brachiosauridae receives support from phylogenetic analysis under both equal and implied weights parsimony, providing the most convincing evidence to date that brachiosaurids dispersed into Asia at some point in their evolutionary history. Consideration of a possibly associated partial dentary (YJDM 00006) from the same site does not impact this conclusion. Several paleobiogeographic scenarios could account for the occurrence of a middle Cretaceous Asian brachiosaurid, including dispersal from either North America or Europe during the Early Cretaceous. These hypotheses can be tested by continued study of excavated specimens from the Longshan locality and future excavation in the Longjing Formation.

Supplemental Information

Supplemental Information 1 Phylogeny data matrix.

Click here for additional data file.

The authors would like to thank Ding Xiaoqing for preparing the specimen, James Tierney for his assistance in proofreading the manuscript, and Joey Stiegler and Ren Xinxin for discussion. Thoughtful comments by editor Andrew Farke and reviewers Philip Mannion and Verónica Díez-Díaz improved an earlier version of this manuscript. We are grateful to the Willi Hennig Society for sponsoring the free distribution of the TNT software.

Institutional abbreviations

CM Carnegie Museum of Natural History, Pittsburg, USA

IVPP Institute of Vertebrate Paleontology and Paleoanthropology, Beijing, China

MB.R. Museum für Naturkunde, Berlin, Germany; MDT-PV, Museo Desiderio Torres-Paleovertebrados, Sarmiento, Chubut, Argentina

PMU Palaeontological Museum, Uppsala, Sweden

USNM National Museum of Natural History, Smithsonian Institution, Washington, DC, USA

YJDM Yanji Dinosaur Museum, Jilin, China

Additional Information and Declarations

Competing Interests

Author Contributions

Field Study Permissions

Data Availability

The authors declare that they have no competing interests.

Chun-Chi Liao conceived and designed the experiments, performed the experiments, analyzed the data, prepared figures and/or tables, authored or reviewed drafts of the paper, and approved the final draft.

Andrew Moore conceived and designed the experiments, performed the experiments, analyzed the data, prepared figures and/or tables, authored or reviewed drafts of the paper, and approved the final draft.

Changzhu Jin performed the experiments, prepared figures and/or tables, and approved the final draft.

Tzu-Ruei Yang performed the experiments, prepared figures and/or tables, authored or reviewed drafts of the paper, and approved the final draft.

Masateru Shibata performed the experiments, prepared figures and/or tables, and approved the final draft.

Feng Jin performed the experiments, prepared figures and/or tables, and approved the final draft.

Bing Wang performed the experiments, prepared figures and/or tables, and approved the final draft.

Dongchun Jin performed the experiments, prepared figures and/or tables, and approved the final draft.

Yu Guo performed the experiments, prepared figures and/or tables, and approved the final draft.

Xing Xu conceived and designed the experiments, performed the experiments, analyzed the data, prepared figures and/or tables, authored or reviewed drafts of the paper, and approved the final draft.

The following information was supplied relating to field study approvals (i.e., approving body and any reference numbers):

Field sites permission was approved by Yanji Paleontological Research Centre (project name: Yanji Dinosaur Fossils Excavation Research Project Cooperation Agreement).

The following information was supplied regarding data availability:

The raw data of CT scan is available at MorphoSource: DOI 10.17602/M2/M361358.

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
