# Peer review of "A possible brachiosaurid (Dinosauria, Sauropoda) from the mid-Cretaceous of northeastern China"

_PeerJ, doi:10.7717/peerj.11957_

## Round 0.1 · original submission · Minor Revisions

Overall, the reviewers' comments are quite positive, and minimal revision is necessary. As you revise the manuscript, I would ask you to pay particularly close attention to the following points:

1) Please add some caveats about the knowledge gaps in maxillary evolution for non-titanosaurian somphospondylans, and how future data may test your hypothesis that this is a brachiosaurid maxilla.

2) Also as suggested by Reviewer 1, add a brief statement on what other sauropod clades currently are identified at the site, and how this might affect your hypothesis. Are other elements consistent with brachiosaurid sauropods? If not brachiosaurid, could the maxilla conceivably be associated with the postcrania?

3) Reviewer 2 requests some clarification on discussion of the antorbital fenestra vs. narial fossa, as outlined in their marked-up document at the relevant part of the description, as well as the potential implications for the phylogenetic analysis. This should be addressed in your revision itself or in the response document.

I look forward to seeing your revised manuscript.

·

Basic reporting

No comment

Experimental design

No comment

Validity of the findings

No comment

Additional comments

Dear authors,

This is an extremely well-written and thorough MS describing a new sauropod specimen from the late Early Cretaceous of China, with an interesting and surprising identification. Based on the available data (both presented here and more broadly), the identification of this specimen as a brachiosaurid is as well-supported as possible. I’ve provided a small number of minor comments on an annotated version of the MS and have one slightly more substantial comment here.

Although I think the MS does a good job of exploring the possible alternative identifications, I think there is one possibility that is not fully covered and I would urge the authors to add some further caveats to their study. Firstly, we know very little about non-titanosaurian somphospondylan maxillary evolution (and arguably essentially nothing if Euhelopus lies outside of Macronaria). As such, it is conceivable that the presence of twisted maxillary teeth characterizes Titanosauriformes more widely (especially if Europasaurus lies outside of this clade). I’d strongly recommend highlighting this knowledge gap as a possible issue with a brachiosaurid identification, essentially saying that new materials of ‘basal’ somphospondylans are needed to ultimately test this, but that current evidence indicates that it’s a unique synapomorphy of Brachiosauridae. The second caveat is what do the rest of the sauropod specimens from the site indicate taxonomically? Is there any clue from (presumably) preliminary observations? Because if you ultimately identify a single somphospondylan taxon from these remains (or find no unequivocal evidence for taxa outside of that clade) and the maxilla is the only anomaly, it will perhaps seem increasingly likely that the maxilla belongs to that taxon instead and the twisted maxillary teeth reflect either convergence or a wider synapomorphy.

I think these caveats will strengthen your paper and somewhat ‘future proof’ it if we do eventually find an unequivocal somphospondylan with twisted maxillary teeth.

Best wishes,

Phil Mannion

·

Basic reporting

The main structure of the manuscript is adequate, the tables provide useful data, and the figures are of good quality. The authors show an impressive knowledge of the previous references, and cite them correctly through the text. The English is professional and the sentences clear, which make the text easy to read and follow.
Generally, the content is really interesting, and provide interesting information about a cranial element found in the Longshan locality (mid-Cretaceous, China), which probably belonged to a brachiosaurid sauropod.

Experimental design

The research question and methods are well described, and properly developed in the manuscript.

Validity of the findings

The results are interesting, and provide new information on the Cretaceous sauropod faunas from China, and also sauropod teeth replacement generation and patterns. The authors also provide an updated phylogenetic analysis, and improve some previous palaeobiogeographical hypotheses, which help to explain the occurrence of a middle Cretaceous Asian brachiosaurid.

Additional comments

Generally, this work is of good quality and interest. However, I would suggest to address two points that could impove the manuscript:
- Although the authors provide references which describe with more detail the geology and location of the site, I would suggest including a map, at least for placing the site.
- I think the authors confused the antorbital fenestra with the narial fossa. I suggest checking this, and adapt the description, comparison and scoring of this feature in the matrix depending on the identification.

---

## Round 0.2 · accepted · Accept

Thank you for your close attention to the reviewer comments.